# Effects of Mechanization and Investments on the Technical Efficiency of Cassava Farms in Cambodia

Tamon Baba [1], Hisako Nomura [2,*], Pao Srean [3], Tha Than [3] and Kasumi Ito [4]

1    College of Letters, Ritsumeikan University, Kyoto 603-8577, Japan; babatmn@fc.ritsumei.ac.jp
2    Faculty of Agriculture, Kyushu University, Fukuoka 819-0395, Japan
3    Faculty of Agriculture and Food Processing, National University of Battambang, National Road No. 5, Sangkat Preaek Preah Sdach, Battambang 021402, Cambodia; srean.pao@nubb.edu.kh (P.S.); than.tha@nubb.edu.kh (T.T.)
4    International Center for Research and Education in Agriculture, Nagoya University, Nagoya 464-8601, Japan; kasumito@agr.nagoya-u.ac.jp
*    Correspondence: hnomura@agr.kyushu-u.ac.jp; Tel.: +81-928024762

**Abstract:** Cassava is one of the most important cash crops in Cambodia. Agricultural mechanization promotes productivity, but overinvestment may disrupt the balance between inputs and outputs. Depending on the production scale, sometimes hiring equipment is considered better than purchasing it. While we can hypothesize that mechanization and investments might be crucial factors of productivity, technical efficiency analysis for estimating their effects has not yet been conducted. Therefore, this study investigates the impact of mechanization and investments on cassava yield and producers' technical efficiency in Cambodia using the Cobb-Douglas stochastic frontier production model. For the study, 205 respondents were randomly selected and interviewed in the Battambang and Pailin provinces in northwestern Cambodia in 2017. Our results show that tractor or truck-hire cost was positively significant, and the cassava uprooting machine-hire cost was negatively significant. The average technical efficiency score of 0.62 indicates that cassava producers can increase their level of technical efficiency. Although cassava production in Cambodia is mechanized and investors are investing, it would be more beneficial to producers if they were provided with financial assistance when uprooting the cassava at the harvest time. Appropriate control of input costs can effectively improve cassava yield, following the implementation of the National Policy on Cassava 2020–2025 by the Royal Government of Cambodia.

**Keywords:** mechanization; stochastic frontier analysis; technical efficiency; production function; cassava; Cambodia

## 1. Introduction

Cassava (*Manihot esculenta* Crantz) has become one of the most important crops in Cambodia due to the response of smallholders to a sharp increase in the external demand for cassava starch as a source of biofuel. As a result, its production in Cambodia has increased rapidly since 2005, reaching 13,512,755 tons in 2019 [1], with an average yield of 27.20 tons per hectare (ha), making it the second-largest producer after Laos in Southeast Asia and the seventh-largest producer worldwide [2]. Additionally, the total cassava harvesting area in the country is 652,531 ha, which is the second largest area after rice, as observed in 2019 [1]. Consequently, the newly introduced cash crop production has changed the income and lives of small-scale producers in Cambodia [3–5].

Cassava production in Cambodia faces many constraints, as it is relatively recent and its production technologies and management practices have not yet been established, and since well-disseminated cassava producers used varieties from Thailand and Vietnam [6]. Based on visual checks done by a descriptor invented by our project, they mostly include improved varieties such as KU 50 and Rayong 7 from Thailand. According

to our survey, cassava production in northwestern Cambodia started around 2011, and Banteay Meanchey, the same northwestern region neighboring Thailand, also witnessed land use transformation to cassava cultivation from 2006 to 2009 [3]. Thus, production technology and management practices, such as the appropriate application of inputs, and production practices, such as ensuring the proper seed density planting, would still need to be learned. Nutrient deficiencies, short crop duration, and high weed density call for improved technologies and management practices [7]. Further, continuous cultivation and inefficient farm management lead to net nutrient removal and the gradual decline of soil fertility [8]. Ou et al. found that seed improvement, soil amendment, and soil erosion control improved cassava yield in the Kampong Cham and Pailin provinces [6]. A more recent study identified the break-even points of the cassava sale price and yield, in addition to the proportion of each cost item to the total revenue in the Battambang and Pailin provinces [8]; however, they did not investigate production efficiency. Increasing the productivity of cassava producers in Cambodia requires estimating its determinants, including the inputs and socio-economic variables that affect cassava production.

One possible chief factor affecting production efficiency is the machinery use and associated costs; uprooting cassava requires labor, as tubers are heavy and well-rooted. Abass et al. showed that the mechanization of cassava processing in Uganda motivated the efficient management and utilization of resources in cassava production [9]. Although agricultural mechanization in Cambodia has been rapidly growing in recent years [10,11], it faces several constraints, requiring the government and other stakeholders to formulate and implement efficient and appropriate measures [11]. Most seedlings' planting and tubers' uprooting processes are still carried out by hand, especially for small-scale producers.

Further, it has been observed that demographic aging and out-migration are prevalent in rural Cambodia. National Route 5 from Battambang provides a convenient route to Siem Reap, the gateway city to the ruins of Angkor, which attracts 2.6 million foreign tourists a year. The city, with its historical architecture, equally allures young people for non-farm employment opportunities from the surrounding provinces. In addition, the people of Battambang and Pailin are more easily able to go to neighboring Thailand to work. Therefore, the study area suffers from a shortage of agricultural labor force. Consequently, many producers rely on hiring machinery at the time of harvest; however, the cost of hiring could increase at the peak time for securing labor to work on the hired machines. Further, "uprooting machine-hire cost" exists only for cassava production. Thus, it might affect cassava production efficiency negatively. While mechanization can promote economic growth through higher technical efficiency, higher yields, and higher net incomes, its impact on cassava production has not yet been sufficiently explored. In addition, the effect of machinery investment on production should also be discussed.

Another determining factor in increasing production efficiency could be farm size. The practice of renting land to expand the farmland for cassava production has been noticed among small-to-medium-scale producers. Deininger et al. observed the positive impact of rental land on producers' income and consequently on agricultural production efficiency [12]. In contrast, Baráth and Fertö found that improving productivity by increasing farm size has limitations unless farms switch technologies, as observed in the case of Hungarian cereal, oilseed, and protein crop-producing farms [13]. Thus, this study investigates whether rental land size or mechanization increases cassava production efficiency in Cambodia.

Technical efficiency analysis has been applied to observe small cassava producers worldwide. For example, there have been studies on Laos [14], Timor [15], Congo [16], Nigeria [17–22], Madagascar [23], Uganda [9], and Thailand [24]. However, the impact of mechanization on cassava yield and technical efficiency is not yet studied. Thus, the study investigates the factors that influence cassava yield, using agricultural machinery as variables. It estimates the producers' technical efficiency regarding cassava production in Battambang and Pailin provinces in Cambodia by employing a Cobb-Douglass stochastic frontier production model.

## 2. Materials and Methods

### 2.1. Study Area and Data Collection

We conducted a series of structured interviews based on a questionnaire on cassava production in two districts of Pailin province and six districts of Battambang province from April to November 2017. These areas are located in northwestern Cambodia, bordering Thailand (Figure 1). Battambang is the largest province, with a total cultivated area of 108,551 ha and 2,620,638 tons of cassava production. Pailin province has 42,110 ha of cultivated area and is the 7th largest producer, with 842,200 tons of cassava production in Cambodia [1]. We studied the situation for one cycle from 2015/2016 (planting) to 2017 (harvesting). The stratified random sampling method based on the cassava plantation area was applied according to the list provided by the village, to categorize the respondents into three groups: those possessing or renting land for cassava production of less than 1 ha, between 1 ha, less than 5 ha, and 5 ha or more. We interviewed household members engaged in cassava farming, and the number of interviewed individual producers was 205:144 from Battambang province and 61 from Pailin province.

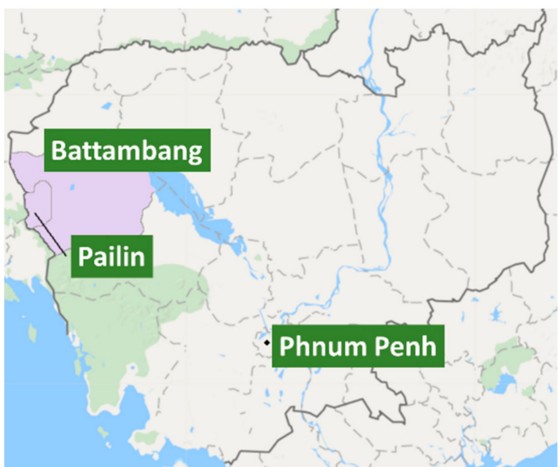

**Figure 1.** Survey areas in Cambodia (created by the authors using Google Maps).

### 2.2. Analytical Framework

We estimated the output-oriented technical efficiency of cassava producers by employing parametric econometric techniques, using the stochastic production frontier approach introduced by Aigner et al. [25] and Meeusen and van Den Broeck [26]. We assume that a cassava producer $i$ produces a vector of a single output, denoted by $Y$, with $Y \in R_+^M$ using inputs $X$. The stochastic production frontier function of the $i$th cassava producer is defined as follows:

$$Y_i = f(X_i, \alpha) \exp(\varepsilon_i), \ i = 1, \dots, N, \tag{1}$$

where all the producers are indexed with a subscript $i$; $Y_i$ denotes the fresh cassava tuber yield level, and $X_i$ is a vector of inputs. For algebraically deriving the cost frontier [27], $\alpha$ is considered the elasticity of input factors, and $\varepsilon_i$ is the composed error term, which is equal to $v_i - u_i$. The term $v_i$ is assumed to be an independently and identically distributed, two-sided, normally distributed random error ($v \sim N[0, \sigma_v^2]$), independent of $u_i$. It represents the stochastic effects outside the producer's control, such as weather, natural disasters, luck, measurement errors, and other statistical noise. The term $u_i$ is a non-negative random error term, which is independently and identically distributed as truncations at zero in the normal distribution, with a mean $-z_i\delta$ and variance $\sigma_u^2$ ($|N(-z_i\delta, \sigma_u^2)|$). It represents the technical inefficiency of the farm. The likelihood function is expressed in terms of the variance parameters $\sigma_v^2 + \sigma^2$ and $\gamma \equiv \sigma^2/\sigma_v^2$, where $0 \leq \gamma \leq 1$.

The following function defines the output-oriented technical efficiency of production for the $i$th cassava producer:

$$TE = \exp(-u_i) = \exp(-z_i\delta - W_i) \tag{2}$$

The technical inefficiency model is thus related to a vector of farm-specific managerial and household characteristics subject to statistical error [28], and can be expressed as follows:

$$u_i = z_i\delta + W_i \geq 0, \tag{3}$$

where $z_i$ is the farm-specific managerial and household characteristics, and the error and $W_i$ are random variables with a normal distribution, such that of $W_i \sim N(0, \sigma_w^2)$.

Since $u_i \geq 0$, $W_i \geq -z_i\delta$, the distribution of $W_i$ is truncated from below at the variable truncation point $-W_i\delta$.

To analyze cassava production and technical efficiency, we collected one output (cassava yield) and production inputs such as labor cost, machine cost, and other production inputs, and socio-economic information such as age, gender, and farming experience with cassava production. Economic conditions in Cambodia point to further inflation. In 2016, the inflation rate was 3.03%: a 1.82% increase from that of 2015 [29]. Although, as we mentioned earlier, there are ample employment opportunities in Thailand or Siem Reap; people from Battambang and Pailin provinces move there to work for higher wages, resulting in the scarcity of labor in cassava production. This contributes to wage increases in cassava production and is included in machine-hire costs. In order to draw comparisons among the hire costs, depreciated machine-purchase costs were also applied to our model as variable costs. Due to this situation, land rent cost can be regarded as a variable. We converted the inputs for cassava production, such as the cost of hiring labor, renting land, and purchasing seedlings, fertilizers, and pesticides into a value per hectare (riels/ha, 4062 riels are almost equivalent to USD 1). In addition to the physical inputs used in production, additional costs incurred by replanting cassava (hereinafter referred to as "replanting cost") were included. When the producers encountered heavy drought or rain, resulting in growth damage during sowing and the early rooting phases, they purchased additional seedlings to replant. It was a usual practice for producers; however, we are not certain if replanting influences output. Thus, the replanting costs were included in the model to examine the production performance under climate change.

To define the model's specifications and significant explanatory variables, we tested the translog function against the Cobb-Douglass function using the likelihood ratio (LR) test, with the null hypothesis being the restricted model or the used Cobb-Douglass function. The test results showed that LR = 311.01 > $\chi^2$ (59, 0.5%) = 0.00. However, the variance inflation factor (VIF) and Breusch-Pagan/Cook-Weisberg test showed multicollinearity (mean value of VIF = 368.29), and our data did not allow us to test the translog function. It could be because the sample size is not sufficiently large to enable us to estimate the high number of coefficients.

The empirical model of the Cobb-Douglas production frontier is defined as follows:

$$lnY_i = \alpha'_0 + \sum_{j=1}^{11} \alpha'_j lnX_{ij} + v'_i - u'_i, \tag{4}$$

where $\alpha_{jk} = \alpha_{kj}$.

The logarithm of the output of a technically efficient producer (using $X_i$ to produce $Y_i^F$) is obtained by setting $u_i$ in Equation (4).

$Y_i$ = cassava yield per hectare (tons); $X_1$ = labor-hire cost per hectare; $X_2$ = field rent cost per hectare; $X_3$ = bunch purchase for the first seedling cost per hectare; $X_4$ = bunch purchase for replanting cost per hectare; $X_5$ = machine fuel cost per hectare; $X_6$ = fertilizer cost per hectare; $X_7$ = other chemical input costs per hectare; $X_8$ = tractor-or-truck-purchase cost per hectare; $X_9$ = sprayer-purchase cost per hectare; $X_{10}$ = tractor or truck-hire cost per hectare; $X_{11}$ = uprooting machine-hire cost per hectare.

$Y_i$: "cassava yield per ha" is the amount of harvested fresh cassava tubers per ha. In some cases, producers sold tubers to silo by themselves, either fresh or dried; in other instances, intermediaries purchased tubers from producers at farmgate. Transported tubers were measured using a large silo-weighing machine, and producers were aware of their yield. Several producers of dried cassava did not know the number of their fresh tubers, while some producers recorded the number of fresh and dried tubers. Based on their records, 1 kg of fresh cassava becomes 0.56 kg of dried cassava, and this ratio was used when the producers were only aware of the number of dried tubers. This calculation agrees with the 0.53–0.57 ratio suggested by Peuo et al. [8]. Similarly, several producers did not observe the total number of tubers but recorded the total profit because intermediaries transported tubers to silos instead of the producers. In this case, we estimated the amount based on the average price of tubers in the market: 140,950 riels/ton for fresh tubers and 418,170 riels/ton for dried tubers.

$X_1$: "labor-hire cost per hectare". We accumulated the total labor-hire cost. Even small-scale producers hire neighbors or others as laborers to plow fields, raise beds, plant cassava cuttings, and apply fertilizers, pesticides, herbicides, as well as for weeding and harvest tubers and stems. It is also possible that neighbors or families support cassava production for free, but we calculated the actual expenditure of hiring labor.

$X_2$: "field rent cost per hectare". Several producers rent fields for cassava production. They pay money to field owners every year.

$X_3$: "bunch purchase for the first planting cost per hectare". Some producers purchase stem bunches for the first planting cycle. Cassava propagates from cutting the remaining cassava stems as seedlings after its tubers are harvested [30]. Therefore, producers keep the harvested stems for the next cycle and do not purchase stems every year. However, they purchase stems when they do not have a sufficient number of stems or want to try new ones. Cassava stems are usually bunched by strings for keeping and handling, with one bunch having 20 stems in general.

$X_4$: "bunch purchase for replanting cost per hectare". Some producers purchase bunches of stems as seedlings for replanting. Cassava fields are easily affected by floods, droughts, pests, and diseases, resulting in the death of stems. In this case, producers remove dead stems, purchase additional stems, and plant them.

$X_5$: "machine fuel cost per hectare". Some producers purchase fuels for machines such as tractors, trucks, sprayers, motorbikes, and cars. We enquired about the fuel cost for these machines used for cassava production.

$X_6$: "fertilizer cost per hectare". Producers use fertilizer to promote cassava's growth. This cost includes manure, urea, $P_2O_5$ (Phosphorus pentoxide), NPK (Nitrogen, Phosphorus, and Potassium), foliar, accelerator, natural fertilizer, and chemical fertilizer.

$X_7$: "other chemical input cost per hectare". This consists of costs for herbicide, pesticide, and fungicide.

$X_8$: "tractor-or-truck-purchase cost per hectare". Tractors have been introduced for plowing and raising beds using accessories, and for transporting tubers, stems, and inputs such as fertilizers. Additionally, they are used for transportation and are commonly purchased by small-scale producers. To estimate machinery-purchase costs, we applied depreciation. To the best of our knowledge, there is no method for calculating depreciation in Cambodia. Therefore, we follow the rules set out by the National Tax Agency in Japan. It is based on a straight-line method rather than the declining-balance method, with tractors and other agricultural equipment having a use life of seven years. If the producers did not remember the purchase price or year of purchasing machinery, we assumed that the machinery was purchased more than seven years ago and, thus, was not included in the depreciation calculation. In the case of second-hand machinery, depreciation was calculated using the straight-line method because of the difficulty in determining its exact value.

Further, when producers used these machines for other crops, the cost was calculated by multiplying the ratio of the cassava field size to the total field size. Although several producers had motorbikes and cars for transporting stem and tubers, they used these

machines more frequently in their daily lives. Therefore, we did not include the cost of these machines in our analysis; however, the fuel costs related to cassava production were added to $X_5$.

$X_9$: "sprayer-purchase cost per hectare". Sprayers are used for applying liquid chemicals, such as herbicides and pesticides. Similar to $X_{11}$, we calculated its cost following the depreciation method by multiplying the ratio of the cassava field size to the total field size.

$X_{10}$: "tractor or truck-hire cost per hectare". Producers hire tractors or trucks for cassava production. Although this cost can include the cost of hired labor for driving and fuel costs, we did not separate them.

$X_{11}$: "uprooting machine-hire cost per hectare". Uprooting machines can be hired for harvesting tubers. Thus, it can include hired labor costs for driving and fuel costs, similar to other hired machines.

Then,

$$u_i' = \delta_0' + \sum_{d=1}^{5} \delta_d' Z_{id} + W_i', \tag{5}$$

where $Z_1$ = age; $Z_2$ = cassava-farming experience, which indicates the number of times the producer has planted or harvested cassava; $Z_3$ = the number of cassava-farming family members, that is, the number of family members participating in cassava production; $Z_4$ = ratio of cassava field size to total field size: the cassava production field (ha) divided by the size of the total agricultural production field (ha), which the producer is cultivating. $Z_5$: "cassava planting density" is the number of plants per m$^2$.

## 3. Results

### 3.1. Parameter Estimates of the Stochastic Frontier Production Function

Table 1 shows the socio-economic characteristics of 205 respondents. The variables denote the factors included in the model, and the mean and standard deviation (St. Dev.); minimum and maximum values of the variables have been described. The average age of the respondents is 49.00 years old, with a wide range, from 21 to 78 years. On average, the respondents have undergone 4.08 cycles of cassava production. Thus, they have limited experience in cassava production.

**Table 1.** Socio-economic characteristics of respondents in Battambang and Pailin provinces.

| Variables | Definition | Unit | Mean | St. Dev. | Min. | Max. |
|---|---|---|---|---|---|---|
| *Production function* | | | | | | |
| yield | Yield | Tons per ha | 17.56 | 9.11 | 0.50 | 52.63 |
| laborhire | Labor-hire cost | Riel per ha | 491,924.95 | 446,503.15 | 0.00 | 2,688,889.00 |
| rentland | Field rent cost | Riel per ha | 100,025.45 | 274,794.95 | 0.00 | 1,380,000.00 |
| seeding | First seedling cost | Riel per ha | 153,892.12 | 382,999.80 | 0.00 | 2,875,000.00 |
| seedreplant | Replanting seedling cost | Riel per ha | 39,978.05 | 111,603.78 | 0.00 | 871,212.13 |
| fuel | Machine fuel cost | Riel per ha | 79,854.05 | 90,109.79 | 0.00 | 477,400.00 |
| fertilizer | Fertilizer cost | Riel per ha | 104,713.20 | 151,631.70 | 0.00 | 920,000.00 |
| otherchem | Other chemical input costs | Riel per ha | 186,183.50 | 146,758.70 | 0.00 | 825,000.00 |
| tractor | Tractor or truck-purchase cost | Riel per ha | 201,427.07 | 649,136.56 | 0.00 | 7,085,715.00 |
| sprayer | Sprayer-purchase cost | Riel per ha | 9413.91 | 23,893.60 | 0.00 | 289,916.00 |
| tractorhireha | Tractor or truck-hire cost | Riel per ha | 501,103.47 | 312,768.86 | 0.00 | 1,467,500.00 |
| uprootinghire | Uprooting machine-hire cost | Riel per ha | 78,940.14 | 105,360.10 | 0.00 | 410,714.28 |
| *Technical efficiency function* | | | | | | |
| age | Age | Years | 49.00 | 13.29 | 21 | 78 |
| cafarmingtimes | Cassava-farming experience | Times | 4.08 | 3.03 | 0.00 | 16.00 |
| #offarmmem | Number of cassava-farming family members | Persons | 2.55 | 1.06 | 0.00 | 6.00 |
| fieldratio | Ratio of cassava field size to total field size | Cassava farm size (ha)/total farm size (ha) | 0.73 | 0.30 | 0.09 | 1.00 |
| density | Density of seed planting | Plants/ha/10,000 | 2.51 | 0.88 | 0.83 | 6.66 |

The number of family members is 4.91 on average, varying from 1 (1 (0.49%) case) to 11 (2 (0.98%) cases) members per family, with 2.55 family members being engaged in cassava production. On average, a cassava field constitutes 72.65% of the total agricultural field. Most of them are produced under mono-cropping, but sometimes under mixed-cropping, which can include longan, mango, and peanuts. Out of 205 producers, 184 producers are mono-cropping cassava. While inputs are used for other crops, the mixed-cropping farm number is limited. However, we asked the producers to separate the input costs to their best knowledge. Thus, we added the variable relating to the ratio of cassava field size to total field size to control for the mixed-cropping producers who might be using inputs for cassava as well as for other crops. In addition to hectares, producers use Thailand unit *rai*, which is equivalent to one-sixth of one hectare. Therefore, we converted *rai* to hectares in this study using this ratio.

In Cambodia, cassava cuttings are planted vertically on the ground. According to our survey, the distance between plants in the row is 0.39 m on average, with a range of 0.20–0.80 m; the distance between plants in two rows is 1.14 m on average, with a range of 0.50–1.50 m. Thus, the planted density varied to about 25,300 plants per ha on average, with a wide range of 8333 to 66,667 plants per hectare. This indicates that Cambodian producers are planting a priori. While the ideal planting density depends on the varieties and soil and is difficult to determine, the Bureau of Agricultural Economic Research in Thailand suggests that cassava should be grown at a density of 10,000 plants per hectare, with a range of 6889 to 15,625 plants, according to the soil fertility [31].

Various costs are involved in cassava production. Owing to shared borders with Thailand, producers in Battambang and Pailin provinces use Thai baht together with Cambodian riel, while purchasing fertilizers, chemicals, and stems from Thailand. We converted Thai baht to Cambodian riels for 115 riels per 1 baht, following the exchange rate. The rental cost for cassava fields was 100,025.45 riels (approximately USD 24.62) per hectare. The cost of bunches was 39,978.05 and 153,892.12 riels (approximately USD 9.84 to 37.88) per hectare for first planting and replanting, respectively. Even small-scale producers need to hire laborers for plowing fields, raising beds, planting cassava cuttings, and applying fertilizers, pesticides, and herbicides, as well as for weeding and harvesting stems and tubers. On average, it costs 491,924.95 riels (approximately USD 121.09) per hectare.

A total of 135 (65.85%) respondents purchased fertilizers for cassava production, including manure, urea, $P_2O_5$, NPK, foliar, accelerators, organic fertilizers, and other chemical fertilizers. Our data show fertilizers cost 104,713.20 riels (approximately USD 25.78) per hectare. Additionally, other chemical inputs such as herbicides, pesticides, and fungicides were purchased by 194 (94.63%) producers. On average, these cost 186,183.50 riels (approximately USD 45.84) per hectare.

Machinery is becoming an essential tool for cassava production. Following the depreciation calculation method, 150 (73.17%) producers spent 201,427.07 riels (approximately USD 49.59) and 14,885.93 riels (approximately USD 3.66) per hectare on purchasing machines (e.g., tractors and trucks) and sprayers, respectively. While the number of producers who bought tractors or trucks was 69 (33.66%), the cost of 20 cases became zero after depreciation. The machine fuel cost was 79,845.05 riels (approximately USD 19.65) per hectare. Further, 188 producers (91.71%) borrowed machines for cassava production. For hiring tractors or tracks, the mean cost is 501,103.47 riels (approximately USD 123.35) per hectare, while the mean cost for hiring uprooting machines is 78,940.14 riels (approximately USD 19.43) per hectare.

The average cassava yield was 17.56 tons per hectare, ranging from 0.50 tons to 52.63 tons per hectare. This is not very different from the results by MAFF, which estimated the yield in Cambodia to be 20.71 tons per hectare [1].

*3.2. Estimated Production Function and Production Technical Inefficiency Model*

Table 2 presents the estimated parameters for the frontier production function model of cassava yield per hectare. Tractor-or-truck-hire cost positively influences the production

at the 10% significance level. In contrast, uprooting machine cost is negative and significant at the 5% level. The result indicates that producers tend to spend a significant amount of money hiring uprooting machines.

**Table 2.** Estimated parameters for the cassava yield per hectare using the stochastic frontier production function.

| Variable | Coefficient | | Std. Dev. |
|---|---|---|---|
| Stochastic frontier production function | | | |
| Dep. var.: the cassava yield per ha | | | |
| lnlaborhire | 0.004 | | (0.005) |
| lnrentland | −0.006 | | (0.005) |
| lnseedling | −0.002 | | (0.004) |
| lnseedreplant | 0.002 | | (0.006) |
| lnfuel | 0.005 | | (0.007) |
| lnfertilizer | 0.005 | | (0.005) |
| lnotherchem | −0.002 | | (0.009) |
| lntractor | 0.002 | | (0.005) |
| lnsprayer | −0.004 | | (0.005) |
| lntractorhire | 0.011 | * | (0.0064) |
| lnuprootinghire | −0.016 | *** | (0.004) |
| Constant | 3.008 | *** | (0.184) |
| Production technical inefficiency model | | | |
| Dep. var.: technical production inefficiency $(u'_i)$ | | | |
| age | −0.063 | | (0.085) |
| cafarmingtimes | 0.180 | | (0.204) |
| #offarmmem | 0.290 | | (0.385) |
| fieldratio | −0.722 | | (1.624) |
| density | −0.214 | | (0.560) |
| Constant | 1.168 | | (1.670) |
| Usigma | | | |
| Constant | −0.887 | | (1.403) |
| Vsigma | | | |
| Constant | −2.583 | *** | (0.592) |
| $E(\sigma_u)$ | 1.179 | *** | [1.137–1.220] |
| $\sigma_v$ | 0.275 | *** | (0.081) |
| LL | −159.385 | | |
| n | 205 | | |

ln is the natural logarithm; standard errors are in parentheses; significance is at *** $p < 0.01$, and * $p < 0.10$, respectively.

The lower half of Table 2 presents the estimated parameters for the technical inefficiency of cassava yield. The variables that were expected to be negative (i.e., efficient), such as age, cassava-farming experience, the number of cassava-farming family members, and the ratio of the cassava field size to the total field size, are not significant. Although further investigation of the multiple outputs is necessary, the tendency shows the higher the ratio, the higher the technical efficiency.

The mean and minimum and maximum technical efficiency scores are presented in Table 3. The frequency distribution of the technical efficiency indices by province is illustrated in Figure 2, showing a wider distribution of technical efficiency through each level of efficiency scores by province. The technical efficiency score is 0.62 for all respondents. The mean technical efficiency scores of 0.60 and 0.67 range from 0.02 to 0.92 and from 0.16 to 0.90 for Battambang and Pailin, respectively. This indicates that producers' productivity in Pailin is higher than that of Battambang. In addition, these results suggest that the inputs used could be reduced by approximately 38% ((1 − 0.62) × 100) for both provinces, and 40% ((1 − 0.60) × 100) and 33% ((1 − 0.67) × 100) for Battambang and

Pailin, without decreasing the current output level. This means that a producer's technical efficiency could be improved to increase the gross margin of producers.

**Table 3.** Technical efficiency scores summary.

|  | Obs | Mean | Std. | Min. | Max. |
|---|---|---|---|---|---|
| Total | 205 | 0.624 | 0.199 | 0.021 | 0.921 |
| Battambang | 144 | 0.604 | 0.202 | 0.021 | 0.921 |
| Pailin | 61 | 0.671 | 0.186 | 0.162 | 0.899 |

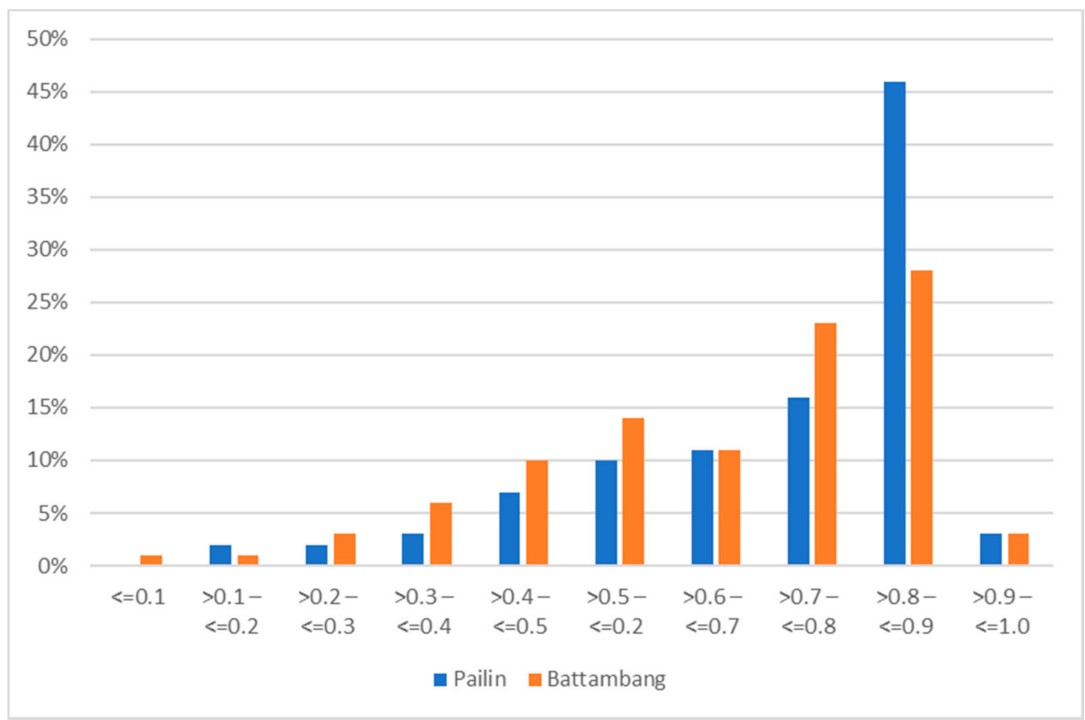

**Figure 2.** Frequency distribution of technical efficiency.

Further, this score also implies that productivity could be increased by 65.52% ((0.60 − 1)/0.60 × 100) and 49.00% ((0.67 − 1)/0.67 × 100), with full efficiency improvement for Battambang and Pailin, respectively. Moreover, the average technically efficient producers could reduce their cost by 34.40% ((1 − (0.60/0.92)) × 100) and 25.39% ((1 − (0.67/0.90)) × 100) in Battambang and Pailin, respectively. Finally, the most technically inefficient producers could save costs by 97.75% ((1 − (0.02/0.92)) × 100) and 81.98% ((1 − (0.16/0.90)) × 100), respectively, if they achieved the maximum technical efficiency level of their counterparts.

## 4. Discussion

The tractor-or-truck-hire cost was positive and significant at the 10% level, with 163 (79.51%) producers hiring them for 399,766.46 riels (approximately USD 98.33) per hectare on average, with a wide range of 9000 riels to 747,500 riels (approximately USD 2.22 to 184) per hectare. The tractor-or-truck-purchase cost was positive but not significant. After depreciation, 49 (23.90%) producers incurred a cost of 842,705.07 riels (approximately USD 207.28) per hectare for the repayment of tractors or trucks for seven years. The uprooting machine-hire cost was negative and significant at the 10% level. A total of 77 (37.56%) producers hired them for 210,165.30 riels (approximately USD 51.70) per hectare. From the above, we can conclude that some producers started purchasing tractors or trucks because hiring tractors or trucks with their drivers was expensive, especially during the peak season, while hiring a tractor or truck was technically efficient. However, no interviewed producer

purchased an uprooting machine; some of the producers hired uprooting machines at high costs. Therefore, we could argue that productivity improves by increasing farm size by way of renting more land; to a certain extent, mechanization is taking place in Cambodia to support productivity improvement. However, many producers have no choice but to hire tracks and uprooting machinery. Therefore, the provision of preferential loan policies targeting small-scale producers' cassava cooperatives or villages could facilitate the introduction of uprooting machines that could reduce the cost of hiring of individuals.

The number of stems, which should be in proportion to the cost of purchasing stems, was reported as a negatively significant variable by Muhaimin [15] and Olurotimi et al., [20] and as a positively significant variable by Ironkwe et al. [19] and Adebayo et al. [21]. In our case, we separated the first seedling and the replanting seedling cost, as we wanted to examine how many the producers must replant (fill the gap in the field) in the case of drought or flooding. In addition, we observed more young stems drying out due to a lack of rain after planting on our study site. However, our study found that the first seedling cost and the replanting seedling cost were negative, and both did not significantly affect productivity.

Furthermore, the amount of fertilizers and chemicals used were as per previous studies. The fertilizer amount was reported as a positively significant variable by Adebayo et al. [21], Ironkwe et al. [19], and Sanusi et al. [22]. Further, the amount of chemicals used is a negatively significant variable, according to Sanusi et al. [22]. In contrast, according to Adebayo et al., herbicides and fertilizers were negatively significant variables in their study areas [21]. In our study, we collected their cost information; while they were not significant, fertilizer and other chemical input costs were negative. This indicates that cassava producers tend to apply these inputs without reflecting on whether they are too expensive and negatively affect productivity, purely based on their experience or knowledge acquired from others, which has not been scientifically proven. Thus, they spend significant amounts of money on such inputs, which is not required.

Finally, it was possible to maximize the technical efficiency of cassava yield by approximately 38%. This means that producers can reduce inputs, particularly the uprooting machine-hire cost, while maintaining the same yield. Our study revealed a technical efficiency score of 0.62 for the cassava production in Cambodia. In comparison to other studies, it was recorded as 0.56 in Laos [14], 0.94 in Timor [15], 0.27 in Congo [16], 0.77 for male producers, 0.74 for female producers [19], 0.83 for innovation platform members and 0.73 for non-members [20], 0.66 in Nigeria [21], 0.79 in Madagascar [23], and 0.61 in Thailand [24]. Further, this score is also similar to the technical efficiency scores of rice production in Cambodia: 0.74 [32] and 0.53 [33]. Although the models and variables differ across studies, we can suppose that cassava production in Cambodia is at almost the same state as other countries or as rice production in Cambodia. The unique point is the significant negative factor, which was uprooting-hired machine, and how it affects the balance between inputs and output.

Overall, the results show the general trends of factors that are contributing to production efficiency in Cambodia at present. Since the cassava cultivation experience of Cambodian producers is limited, standard or traditional production methods may not exist; small-scale cassava producers are growing cassava in their own style. Therefore, we could incorporate our findings into producers' training and workshop materials. In other words, there are numerous possibilities for improving the cassava production in Cambodia.

## 5. Conclusions

This study investigated the determinants that influence the yield of cassava producers in Cambodia using a stochastic frontier model. We used a sample of 205 small-scale cassava producers cultivating cassava fields of 3.73 ha on average, with limited cassava production experience (4 cycles) in the Battambang and Pailin provinces in northwestern Cambodia.

Our results showed that tractor-or-truck-hire cost is a significantly positive variable. Further, uprooting machine-hire costs is negatively significant. Considering that the mecha-

nization of the agricultural sector in Cambodia has improved rapidly and truck-or-tractor-hire cost has positively affected cassava production, it is noted that the cassava tuber uprooting machine hires cause imbalance in inputs and output. One of the strategies for cassava producers can be to hire or purchase uprooting machines at lower costs. The provincial or regional financial support for uprooting activity can help reduce the negative impact.

Further, many producers rely on hiring machinery during the entire production period; however, the cost of hiring could rise at the peak time for securing labor along with the hired machines. The study indicates that labor and fuel costs could influence the potential trade-off between the technological advantages of mechanization and the cost disadvantages in employing machinery during the peak period. Further, we should be aware that truck or tractor-hire costs and uprooting machine-hire costs are significantly influenced by labor and fuel costs for cassava production. Thus, depending on the labor and fuel costs, these determinants might affect cassava production efficiency negatively. While mechanization can promote economic growth through higher technical efficiency, higher yields, and higher net incomes, its impact on cassava production has not yet been sufficiently explored.

We estimated the technical efficiency score of cassava yield to be 0.62, indicating that cassava producers have many opportunities to increase their technical efficiency level. Thus, the overall production efficiency could improve if the information on production technologies and management practices is accumulated and shared with the individual farmers.

To summarize, efficient production methods, including calculating and saving costs for cassava yield, have not been established in Cambodia because cassava production is a recent phenomenon there. Based on output-oriented efficiency, producers have opportunities to increase technical efficiency. Furthermore, based on our findings, we can suggest that cost management can help producers (we had some workshops already in Cambodia), and financial support for uprooting machines might be helpful. Therefore, the appropriate control of input costs, such as uprooting machine-hire costs, could be effective in improving cassava yield. Production can be increased if the proposals for producers, consistent with the National Policy on Cassava 2020–2025 by the Royal Government of Cambodia [34], are achieved.

**Author Contributions:** Conceptualization, H.N. and K.I.; methodology, H.N.; software, H.N. and T.B.; validation, H.N., P.S. and K.I.; formal analysis, T.B. and H.N.; investigation, T.T., T.B., H.N. and K.I.; resources, P.S.; data curation, P.S., T.B., K.I. and H.N.; writing—original draft preparation, T.B.; writing—review and editing, H.N.; supervision, H.N.; project administration, Nobuyuki Iseri; funding acquisition, K.I. All authors have read and agreed to the published version of the manuscript.

**Funding:** This research was supported by the Science and Technology Research Partnership for Sustainable Development (SATREPS), in collaboration with the Japan Science and Technology Agency (JST, JPMJSA1508) and Japan International Cooperation Agency (JICA).

**Institutional Review Board Statement:** Not applicable.

**Informed Consent Statement:** Informed consent was obtained from all subjects involved in the study and we explain how the data was anonymously handled and managed.

**Data Availability Statement:** Not applicable.

**Conflicts of Interest:** The funders had no role in the design of the study, collection, analyses, interpretation of data in the writing of the manuscript, decision to publish the results.

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
