# Peer review of "Effects of Mechanization and Investments on the Technical Efficiency of Cassava Farms in Cambodia"

_agriculture, doi:10.3390/agriculture12040441_

Round 1
Reviewer 1 Report
Please see the attachment.

Author Response
Dear Reviewer 1,
Thank you very much for your insightful review. Please see the attachment.
Sincerely yours,
Hisako Nomura

Reviewer 2 Report
- In my opinion, the title of the article is not the best in relation to its content. The title suggests that the main attention will be focused on the analysis of the impact of mechanization (more broadly, outlays on investments in the machinery park) on the technical efficiency of cassava cultivation. Meanwhile, mechanization is one of the many factors included in the analysis (as an input variable).
- There is no well-defined research problem and research goal. We do not find this content in the abstract or in the Introduction. Generally the abstract is weak.
- Some of the inputs adopted for the analysis of the technical efficiency of cassava production are quite controversial. It can be understood that fertilizer or herbicide inputs are variable inputs directly affecting yields, but the field rent cost per ha is a fixed input for a given farm and is not related to yield.
- What is the level of mechanization in Cambodia's cassava production in general? The content of the article shows that it is rather low, but it is not well described.
- The conclusions of an application nature are quite general, they relate mainly to the launch of a program of preferential loans for financing capital expenditure (mechanization) of cassava production. Are these the only conclusions from this analysis?
- What conclusions from the research can be directed to cassava producers in Cambodia?
Author Response
Dear Reviewer 2,
Thank you very much for your insightful review. Please see the attachment.
Sincerely yours,
Hisako Nomura

Reviewer 3 Report
This paper analyses the determinants of cassava farms’ efficiency in Cambodia, using the translog stochastic frontier production model. This issue is relevant particularly for developing countries where cassava plays a major role in food security. The paper is well organized, clear, and insightful. The methodological options are well-founded, and the results are logically presented. I think, however, that some minor improvements could be made in the Material and Methods and Results sections to make results easier to understand.
Material and Methods:
- In the Material and Methods section, information regarding data collection is missing. Did the authors used secondary sources of statistical data, or did they perform a survey? If so, some additional information must be given: type of survey and main questions asked to participants; how many participants were involved and how were they selected; how participants were distributed between Battambang and Pailin provinces; when was the survey performed; …
- I think that the authors used cost as a proxy for agricultural intensification level (level of use of production inputs such as machinery). That should be very well explained because it doesn´t make sense that the truck purchase cost, for instance, is positively related to yield. In fact, if the tractor purchase cost increases, it would be expected that the number of farmers purchasing a tractor would be smaller, negatively influencing their farms’ yield.
Results:
- For readers not familiar with this type of models it is not easy to understand what the values in Table 1 horizontal axis represent. Some explanation should be given in the text to make clearer for general readers the meaning of the results presented between lines 359 and 369.
Additional comments:
- In the abstract, in addition to methodology and results, some background and motivation for the study are missing
- In Table 1, for dichotomic variables, such as gender or farming status, the value presented in the column mean corresponds, in fact, to relative frequency. For accuracy, a note should be put at the end of the table to stress this issue and standard deviations should be omitted.
- The title of Fig. 2 is showed twice, and the horizontal axis is labeled as TE which I assume stands for Technical Efficiency. However, the acronym is not displayed anywhere. Maybe it could be put in the title of the figure.
Author Response
Dear Reviewer 3,
Thank you very much for your insightful review. Please see the attachment.
Sincerely yours,
Hisako Nomura
